# Automatic Modulation Recognition Based on the Optimized Linear Combination of Higher-Order Cumulants

**DOI:** 10.3390/s22197488

**Published:** 2022-10-02

**Authors:** Asad Hussain, Sheraz Alam, Sajjad A. Ghauri, Mubashir Ali, Husnain Raza Sherazi, Adnan Akhunzada, Iram Bibi, Abdullah Gani

**Affiliations:** 1Faculty of Engineering & Computer Sciences, National University of Modern Languages, Islamabad 44000, Pakistan; 2Department of Engineering and Applied Sciences, University of Bergamo, 24129 Bergamo, Italy; 3School of Engineering & Applied Sciences, ISRA University, Islamabad Campus, Islamabad 44000, Pakistan; 4Department of Management, Information and Production Engineering, University of Bergamo, 24129 Bergamo, Italy; 5School of Computing and Engineering, University of West London, London W5 5RF, UK; 6College of Computing and Information Technology, University of Doha for Science and Technology, Doha 24449, Qatar; 7Department of Computer Science, Comsats University, Islamabad 45550, Pakistan; 8Faculty of Computing and Informatics, University Malaysia Sabah, Kota Kinabalu 88400, Malaysia

**Keywords:** modulation recognition, K-nearest neighbor, genetic algorithm, higher-order cumulants

## Abstract

Automatic modulation recognition (AMR) is used in various domains—from general-purpose communication to many military applications—thanks to the growing popularity of the Internet of Things (IoT) and related communication technologies. In this research article, we propose an innovative idea of combining the classical mathematical technique of computing linear combinations (LCs) of cumulants with a genetic algorithm (GA) to create super-cumulants. These super-cumulants are further used to classify five digital modulation schemes on fading channels using the K-nearest neighbor (KNN). Our proposed classifier significantly improves the percentage recognition accuracy at lower SNRs when using smaller sample sizes. A comparison with existing techniques manifests the supremacy of our proposed classifier.

## 1. Introduction

The Internet of Things (IoT) has offered us many new sensor applications that are aimed at making our lives easier [1,2]. The idea of the IoT necessitates the deployment of a large number of low-cost, low-power-consumption, and low- to moderate-range sensors [3]. The IoT can gather data from the surrounding environment and transmit them through a wireless link to a user application for optimum mobility and cost effectiveness [4,5]. Consequently, we have a heterogeneous network with a variety of communication technologies that each serve a variety of applications with varying communication needs, such as their data rate, range, delay tolerance, connection, etc. [6,7].

A typical heterogeneous network consists of different systems and technologies [8,9]. Some of the potential systems and technologies are device-to-device communication [10], cognitive radio communication [11], intelligent transportation [12], unmanned aerial vehicles [13], intelligent reflecting surfaces [14], artificial intelligence [15], and satellite communications [16,17]. In a heterogeneous network environment, IoT devices face some common security concerns, such as privacy, authentication, administration, information storage, and so on, which may lead to quality-of-service (QoS) degradation [18,19]. Automatic modulation recognition (AMR) can be instrumental in decoding unknown signals that may cause interference or privacy invasion [20]. The knowledge about the signal modulation can then be used for various purposes, such as tracking or jamming the communication of malicious users and tracking the signals of desired users, which are very important in spectrum regulations. This is the main reason that AMR has recently been used by many researchers for the IoT [21] and cognitive radio networks [22].

The process of AMR is broadly classified into two approaches, i.e., the likelihood-based decision-theoretic (DT) approach and the feature-based (FB) pattern recognition approach. Both the DT and FB approaches have their fair share of merits and demerits. The DT approach is based on the probability density function (PDF) prediction of the received signal. It is comparatively better in terms of accuracy, but computationally complex and poorly robust to model mismatches [23,24].

On the other hand, the FB approach is a sub-optimal method that relies on statistical features, i.e., higher-order statistical cumulants of the received signal instead of a PDF, thus making it easier to implement and robust to model mismatches, but at the cost of accuracy. Another important advantage of the FB approach is it can differentiate between the different orders of modulation in the same class, such as 16-QAM and 32-QAM, by using appropriate-order cumulants [25,26]. Higher-order cumulant (HOC) behavior in different transformations is essential in evaluating how useful these statistics may be in characterizing the signal. Only the average of the received signal is modified by translations, keeping the variance and all of the HOC intact [27]. Although there is much research available in the field of FB-based AMR, common issues regarding cumulant-based feature extraction are affected by noise at lower SNRs and the number of samples. In fact, to increase classification accuracy, not only better SNR levels, but also a higher-order number of samples are required.

### 1.1. Motivation of the Research

The primary goal of this research project is to make use of the FB technique by examining the synergy between the legacy of meta-heuristic techniques for optimizing the HOCs and the classical mathematical technique of using a linear combination of optimized HOCs to design an efficient classifier.

### 1.2. Contribution of the Research

The contribution of the research work presented in this article is manifold. In this research work, we introduce a unique approach for modulation recognition of five modulation schemes, i.e., BPSK, QPSK, QAM, 16-QAM, and 64-QAM. Instead of using the HOCs directly, a linear combination (LC) of HOCs to create super-cumulants using arbitrary coefficients is used as the feature set. For every two super-cumulants, the coefficients are heuristically computed so that the distance between two modulation schemes is optimized. After that, these optimized coefficients are fed back into the classifier structure, where the distance is calculated using the KNN. From the extensive simulations, significantly higher recognition accuracy was achieved for all of the modulation schemes at lower SNRs. Moreover, for fading channels, there was a considerable improvement in the percentage recognition accuracy at a lower number of samples.

### 1.3. Structure of the Article

The research article is structured as follows: A summary of the existing literature is presented in Section 2. The system model is described in Section 3. A detailed description of our proposed solution based on the genetic algorithm (GA)-assisted linear combination (LC) of higher-order cumulants (HOCs) is discussed in Section 4. Section 5 presents a performance analysis in the form of a discussion on exhaustive simulation results and a comparison with the state of the art. The research article is then concluded in Section 6, and some research directions for extending this work are pointed out.

## 2. Related Work

In the current era, different DL-based solutions have been presented for the security of IoT devices [28,29,30,31,32,33], as well as for modulation classification. A deep-learning-based architecture was used in [34] to propose a modulation classifier using quasi-recurrent neural network (SQRNN) layers for CR-IoT applications. The authors claimed to achieve better classification accuracy, as well as low computational complexity, in comparison with the results with different CNN- and RNN-based classifiers. A realistic generalized CNN-based modulation classifier was presented in [35], which achieved higher robustness compared to traditional automatic modulation classifiers (AMRs). The suggested technique was distinguished by the fact that it was trained on a mixed dataset for extracting common features under various noise conditions.

The authors of [36] proposed a classification system for the recognition of different variants of modulations, i.e., PSK, QAM, DVB-S2, and APSK, by first optimally selecting up to sixth-order features and a radial basis function (RBF) for classification. An approach was proposed in [37] to classify the modulated signals by combining the Wiener filtering method and backpropagation neural networks to mitigate the poor performance of HOCs under noise. In [38], the authors presented a multi-dimensional feature extraction of instantaneous information and HOCs in order to realize modulation recognition. Furthermore, a new characteristic parameter was presented to increase the modulation recognition ability. A tree-shaped multi-layer smooth support vector machine (SVM) classifier based on the feature selection technique was described in detail [39], as was a mixed classification algorithm based on the two new features.

A comprehensive study that combined random erasing and attention methods using a single-layer deep learning LSTM for an AMR framework was presented in [40]. Two random erasing-based data augmentation strategies were also included to improve the model’s generalization capability and robustness. The problem of enhancing the M-FSK signal modulation classification performance in the context of an AWGN was investigated in [41]. In [42], an optimal modulation classification technique was presented, which combined Gabor feature extraction and cuckoo search optimization (CSO). The authors of [43] presented a robust deep-learning-based AMR model for adapting noise variation in a channel. For modulation classification, a modulation recognition cluster network (MRCN) was developed after an SNR estimator (SEN) determined the SNR values of samples. To help with the integration of the SEN and MRCN, a label-smoothing technique was also suggested.

The authors presented a k-sparse auto-encoder-based classifier to reduce the computational complexity of the AMR system. In comparison with a linear SVM, approximate maximum likelihood classification (AMLC) and a Bayesian confidence propagation neural network (BCPNN) were investigated. Varying SNR conditions were explored, and noise-insensitive features based on set theory were used to improve the accuracy [44]. The authors of [45] used deep-learning-based sparse auto-encoders with non-negativity constraints and fourth-order cumulants for modulation classification. In [20], an FB-AMR framework was proposed, which utilized blind channel estimation and maximum likelihood (ML)-based multi-cumulant classification. FB-AMR for an MIMO system was proposed in [46], which used HOCs with a quasi-Newtonian method.

In [47], digital and analog modulation schemes under different SNRs were classified using a random forest. The authors of [48] considered M-PSK and M-QAM for AMR. The effectiveness of features consisted of a generalized autoregressive conditional heteroscedasticity (GARCH) model with a discrete wavelet transform (DWT). In [49], the authors adopted a GP-based approach to produce super-features from the dataset and then improved the performance of the KNN classifier. Two low-complexity classifiers based on order statistics were presented in [50], and LSVM algorithms were investigated for classification. In [51], deep neural networks for AMR were presented, and a stock well transform for underwater acoustic channels was presented in [52]. The SVM classifier was used with the energy entropy of s-transform time–frequency spectrum signals. A high-efficiency classification system using a genetic algorithm with a backpropagation neural network for four statistical features was developed in [53].

In [54], a comparative analysis of different distance methods using KNN classification of five different modulation schemes was presented. The authors showed that the Mahalanobis distance was better in terms of classification accuracy compared to the Minkowski, Euclidean, and correlation methods. The KNN was utilized for feature selection and then detection in [55]. In [56], fourth-conjugate cumulants of the estimated symbols were used to classify the modulation types. In [57], AMR was performed using step-wise regression and hierarchical polynomial classifiers, and HOCs were used to achieve higher accuracy. The authors of [58] used variational mode decomposition (VMD) for AMR when AWGN and non-Gaussian impulsive noise were present. A three-step methodology was adopted for the recognition of QPSK, 16-PSK, 64-PSK, QAM, 16-QAM, and 64-QAM in [59].

In [60], instantaneous features, such as instantaneous amplitude, phase, and frequency parameters, were utilized for AMR. Analog and digital signal classification was reported in [61] by using statistical characterization and an artificial neural network (ANN). In [62], modulation classification based on a constellation diagram with fuzzy logic was presented for QAM signals. A comprehensive review of AMR was presented in [63] that not only compared different types of AMR methods, but also provided the different types of software packages for AMR methods, as well as practical challenges in the implementation process.

## 3. Proposed System Model

The system model of the proposed modulation recognition framework is presented in Figure 1. The expression of the received signal is in Equation (Equation 1): (1)v(t)=h(t)∗x(t)+η(t)
where x(t) and v(t) are the transmitted and received signals, respectively. η(t) is the additive white Gaussian noise (AWGN), and h(t) represents the Rayleigh fading coefficients.

The input signal is modulated, i.e., BPSK, QPSK, QAM, 16-QAM, and 64-QAM, and transmitted over the Rayleigh fading channel in addition to AWGN. The main reason for choosing these modulation formats for the problem under consideration was their application in commonly used wireless technologies. The recognition of modulation formats consisted of three phases:Parameter extraction (HOCs);Super features (optimal weight finder);Recognizer (K-nearest neighbor).

The proposed AMR module is presented in Figure 2. The HOCs were selected as features and represented by ζ. The genetic algorithm (GA) was used to find the optimized weights γ. Super feature cumulants *L* were formed by the linear combination of ζ and γ and are represented as: (2)$$L_{i}=\gamma_{m*1}^{T}.\;\zeta_{m*1}$$
where *L* is be computed for all five modulation scenarios: (BPSK, QPSK, QAM, 16-QAM, 64-QAM). The KNN recognizer computes the Euclidean distance (ED) to decide the modulation format of the received signal.

## 4. Proposed AMR Algorithm

The proposed AMR algorithm mainly consists of three phases:(i)Feature extraction;(ii)Super cumulant feature;(iii)Recognizer.

### 4.1. Feature Extraction

For each of the five modulation techniques, HOCs were derived from the received signal. The cumulants were made up of moments, which can be presented by the general expression shown in Equation (Equation 3), whereas the extracted HOCs were formulated from Equations (Equation 4)–(Equation 13). The features extracted from the received signal are tabulated in Table 1.
(3)$$m_{pq}=E\left[v{\left(t\right)}^{p-q}v^{*}{\left(t\right)}^{q}\right]$$
(4)$$\zeta_{1}=C\left\{v(t),v(t)\right\}=E\left[v^{2}\left(t\right)\right]$$
(5)$$\zeta_{2}=C\left\{v\left(t\right),v^{*}\left(t\right)\right\}=E\left[{\left|v(t)\right|}^{2}\right]$$
(6)$$\zeta_{3}=C\left\{v\left(t\right),v\left(t\right),v\left(t\right),v^{*}\left(t\right)\right\}=m_{40}-3m_{20}m_{21}$$
(7)$$\zeta_{4}=C\left\{v\left(t\right),v\left(t\right),v^{*}\left(t\right),v^{*}\left(t\right)\right\}=m_{42}-|m_{20}{|}^{2}-2m_{21}$$
(8)$$\zeta_{5}=C\left\{v(t),v(t),v(t),v(t),v(t),v(t)\right\}=m_{60}-15m_{20}m_{40}+30m_{20}^{3}$$
(9)$$\zeta_{6}=C\left\{v\left(t\right),v\left(t\right),v\left(t\right),v\left(t\right),v\left(t\right),v^{*}\left(t\right)\right\}=m_{61}-5m_{21}m_{40}-10m_{20}m_{41}+30m_{20}^{2}m_{21}$$
(10)$$\zeta_{7}=C\left\{v\left(t\right),v\left(t\right),v\left(t\right),v\left(t\right),v^{*}\left(t\right),v^{*}\left(t\right)\right\}=m_{62}-6m_{20}m_{42}-8m_{21}m_{41}-30m_{22}^{2}m_{40}+6m_{20}^{2}m_{22}+24m_{21}^{2}m_{22}$$
(11)$$\zeta_{8}=C\left\{v\left(t\right),v\left(t\right),v\left(t\right),v^{*}\left(t\right),v^{*}\left(t\right),v^{*}\left(t\right)\right\}=m_{63}-9m_{21}m_{42}+12m_{21}^{3}-3m_{20}^{2}m_{43}-36m_{22}m_{41}+18m_{20}m_{21}m_{22}$$
(12)$$\zeta_{9}=C\left\{v(t),v(t),v(t),v(t),v(t),v(t),v(t),v(t)\right\}=m_{80}-35m_{40}^{2}-28m_{60}m_{20}+420m_{40}m_{20}^{2}-630m_{20}^{4}$$
(13)$$\zeta_{10}=C\left\{v\left(t\right),v\left(t\right),v\left(t\right),v\left(t\right),v^{*}\left(t\right),v^{*}\left(t\right),v^{*}\left(t\right),v^{*}\left(t\right)\right\}=m_{84}-16\zeta_{8}\zeta_{2}+|\zeta_{3}{|}^{2}-18{\zeta_{4}}^{2}-72\zeta_{4}{\zeta_{2}}^{2}-24{\zeta_{2}}^{4}$$

### 4.2. Super Cumulant Features

In this module, super cumulant features were computed using optimized weights γ from the genetic algorithm (GA). The objective was to maximize the distance between two classes. The linear combination of these constants with the HOCs resulted in super cumulant features. The super cumulant features were computed for each modulation scheme, i.e., BPSK, QPSK, QAM,16QAM, and 64QAM. The linear combination of the super cumulant features was calculated using Equation (Equation 14), as shown below: (14)$$L_{i}=\sum_{k=1}^{M}\gamma_{k}.\;\zeta_{k}$$where $L_{i}$ denotes the linear combination for the *i*th modulation type and *M* is the number of cumulants for each modulation format. In vector form, γ and ζ are:$$\gamma =[\gamma_{1},\gamma_{2},\gamma_{3},\gamma_{4},\gamma_{5},\gamma_{6},\gamma_{7},\gamma_{8},\gamma_{9},\gamma_{10}]$$$$\zeta =[\zeta_{1},\zeta_{2},\zeta_{3},\zeta_{4},\zeta_{5},\zeta_{6},\zeta_{7},\zeta_{8},\zeta_{9},\zeta_{10}]$$

The fitness function for the optimal weight finder using the GA is given in Equation (Equation 15): (15)$$FF=arg_{max}{∥L_{i}-L_{j}∥}^{2}i\neq j$$

The super features for each modulation are presented in Table 2. The algorithm for finding the optimal weights and then the super cumulant features are in Algorithm 1. Algorithm 1 presents a stepwise summary of the computation of the optimized weights using the GA.

**Algorithm 1:** GA-Based Optimal Weight Finder.

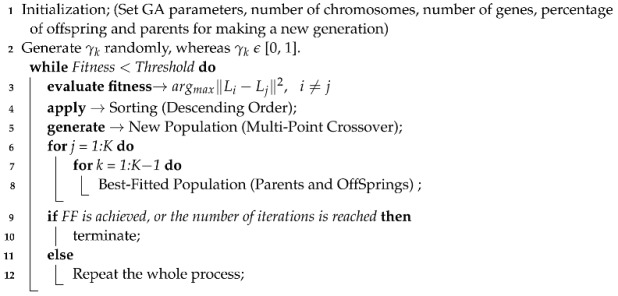



### 4.3. KNN Recognizer

In the literature, the K-nearest neighbor (KNN) recognizer is frequently used to compare the effectiveness of various classifiers. The KNN recognizer kernel is predicated on the calculation of the distance or resemblance between the tested and training samples. Investigation with the KNN recognizer requires no prior information on the data distribution. The data points (nearest neighbors) i.e., K=5, are chosen. All points in each neighborhood are weighted equally, i.e., uniform weights, and from each data point, the Euclidean distance is calculated as shown in Equation (Equation 16); this is the performance metric. The KNN recognizer has the following steps:Load training and test data;super feature cumulants are the dataset.Choose K, i.e., the data points that are closest to it; the chosen value of K is 5 in this research.Perform the following for each data point:
Measure the distance between each row of training data and the test data; the distance is calculated using the Euclidean distance formula, as in Equation (Equation 16):
(16)$$D=\sqrt{\left(L_{i}-L_{i}^{*}\right){\left(L_{i}-L_{i}^{*}\right)}^{*}}$$
where $L_{i}$, the input, is the feature value, and $L_{i}^{*}$ is the test feature value.Distance values are sorted in increasing order.Select the first K rows of the sorted array.The most prevalent class among these rows will now be used to determine the class for each test point.
Stoppage criterion.

The proposed AMR algorithm is shown in Algorithm 2.

**Algorithm 2:** Proposed AMR Algorithm.

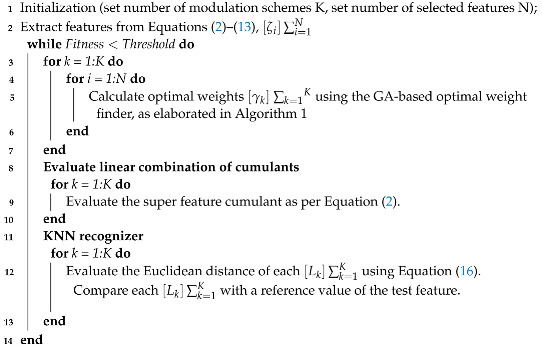



## 5. Simulation Results and Analysis

In this section, detailed simulations are carried out to validate the performance of the proposed super-cumulant-feature-based AMR recognizer. The simulations were performed in MATLAB R2020, on a Windows 10 professional platform with an i5 core processor (sixth generation) and 16 GB of RAM. The merit of the problem is the percentage recognition accuracy (PRA). A comprehensive analysis of our proposed AMR approach and its comparison with existing state-of-the-art techniques are also presented in this section. The modulation of interest for recognition is BPSK, QPSK, QAM, 16-QAM, and 64-QAM. The performance is also compared on a fading channel, i.e., a Rayleigh fading channel with different numbers of samples and different SNRs. The simulation parameters are shown in Table 3.

### 5.1. Performance Analysis on the AWGN Channel

Table 4, shows the percentage recognition accuracy (PRA) on the AWGN channel with different SNRs and numbers of samples for the considered modulation formats. From Table 4, it can be observed that, for all modulation formats, the PRA was increased by increasing the number of samples and the SNR. The proposed recognizer was able to recognize all of the modulation formats with higher accuracy at lower SNRs and even with smaller numbers of samples. At an SNR of 0 dB, the proposed AMR provided 100% PRA with 2048 and 4096 samples, while it provided around 90% with 512 and 1024 samples. The average PRA for the BPSK, QPSK, QAM, 16-QAM, and 64-QAM was 97.5%, 99.3%, 95%, 99.8%, and 99.5%, respectively, with an SNR of 0 dB.

### 5.2. Performance Analysis on the Rayleigh Fading Channel

To further analyze the performance of our proposed AMR approach for the worst-case scenario, i.e., a Rayleigh fading channel, the same set of simulations were repeated, and the results are presented in Table 5. A behavior of the AMR algorithm similar to that in the case of the AWGN channel was observed, showing an increase in PRA at higher SNRs and with larger numbers of samples. The PRA, however, was greatly reduced owing to the absence of the dominant path (non-line-of-sight scenario). Nevertheless, our proposed AMR approach showed good accuracy for all of the combinations of different SNRs and the sample size. The lowest PRA of 80% at an SNR of 0 dB was observed for the QAM signal, and it increased to 99.5% by increasing the number of samples to 4096. The average PRA for BPSK, QPSK, QAM, 16-QAM, and 64-QAM was 95.9%, 97%, 91%, 94.25%, and 96%, respectively, at an SNR of 0 dB.

### 5.3. Performance Comparison on the AWGN and Rayleigh Fading Channels

Figure 3, Figure 4, Figure 5, Figure 6 and Figure 7 show a comparison of the the performance of the proposed AMR approach on the AWGN and Rayleigh fading channels for the BPSK, QPSK, QAM, 16-QAM, and 64-QAM modulated signals at lower SNR values and with varying numbers of samples. Figure 3 illustrates the comparison of the PRA for the BPSK modulated signals at an SNR of 0 dB with different numbers of samples. The bar graph shows that, for every number of samples considered, the PRA for the AWGN was much better than that for the Rayleigh fading channel model. The lowest accuracy was 92%, which was for the Rayleigh fading case when the number of samples was 512 at an SNR of 0 dB.

Figure 4 shows the percentage recognition accuracy for the QPSK modulated signals. With 512 samples, the lowest accuracy of 94% was achieved for the QPSK signals, which was better than the BPSK signal recognition. The super cumulant feature for QPSK was optimized at lower SNRs and with fewer samples.

The comparison of the percentage recognition accuracy for the QAM modulated signals is shown in Figure 5. As shown in Figure 5, the PRA for the Rayleigh fading channel scenario was less than 90% and approximately 100% with 512 and 4096 samples, as compared to the AWGN channel, in which the PRA was approximately 100% with 2048 samples. Similar results are shown in Figure 6 and Figure 7 for the 16-QAM and 64-QAM modulated signals. The PRA was 100% with 4096 samples for the case of the Rayleigh fading channel and for the AWGN scenario.

As can be seen from the results shown in Table 3 and Table 4 and Figure 3, Figure 4, Figure 5, Figure 6 and Figure 7, the proposed AMR approach gave better results for smaller sample sizes at lower SNRs, which was the main motivation behind this research. The utilization of a novel concept of the linear combination of the HOCs, rather than the use of the GA-optimized cumulants, actually caused the samples to be far apart in the decision regions, thus making it much easier to recognize the different modulation samples more accurately.

### 5.4. Comparison with Existing State-of-the-Art Techniques

A comparison of the PRA of the proposed AMR approach with that of existing state-of-art techniques is quantitatively evaluated in Table 6. As seen from Table 6, for [60], the recognition accuracy for BPSK and QPSK was very low at an SNR of 5 dB, while in comparison, our proposed AMR approach achieved 100% at an SNR of 0 dB. Similarly, for [64], the Gabor filter network was used to classify QAM, 16-QAM, and 64-QAM at an SNR of 5 dB with a PRA of 72.35%, 71.94%, and 69.96%. For the same set of modulation schemes, SNR, and number of samples, our proposed AMR approach gave almost 100% accuracy. In our previous work [54], although we achieved higher accuracy for QAM, 16-QAM, and 64-QAM at an SNR of 5 dB, almost 100% accuracy was achieved for all of the considered modulation schemes at an SNR of 0 dB. For the AWGN comparison, as shown in Table 6, the proposed AMR outperformed the others for the all of the modulation formats, even at lower SNRs and numbers of samples.

The PRA of the proposed AMR approach was compared with that of [54] in the presence of a Rayleigh fading channel, as shown in Table 7. The remarkable improvement in the performance of the proposed AMR approach can be seen in Table 7. The PRA was compared for 512 and 1024 samples at SNRs of 0 and 5 dB. In [54], the authors only used the cumulants as a feature set and the KNN as a classifier, but in the proposed AMR approach, the super cumulant features were optimized by using the GA, which showed supremacy in terms of PRA.

## 6. Conclusions

This paper presents modulation recognition using a genetic-algorithm-assisted linear combination of higher-order cumulants. The idea was to reduce the number of features for all modulation schemes and to distinguish between them with a minimal number of comparisons. The AMR approach was divided into three phases: feature extraction, GA-based optimal weight finding for the computation of super cumulant features, and a KNN recognizer. The proposed AMR method gave a better and comparable PRA at lower SNRs and even for lower numbers of samples in comparison with existing methods for the AWGN and Rayleigh channels. In future work, our proposed recognizer can be used for different modulation schemes while considering different communication scenarios. The LC approach can be combined with heuristic algorithms other than the GA to explore further improvements. Similarly, one can explore a deep neural network for recognition instead of the KNN. 

## Figures and Tables

**Figure 1 sensors-22-07488-f001:**
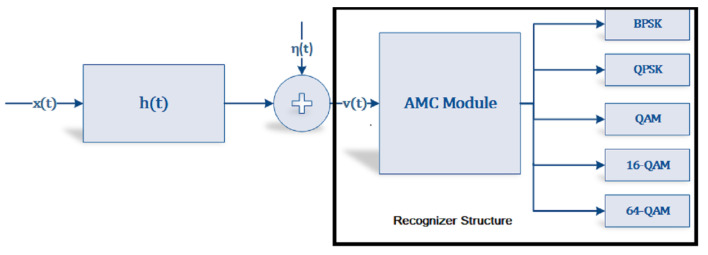
Proposed system model.

**Figure 2 sensors-22-07488-f002:**
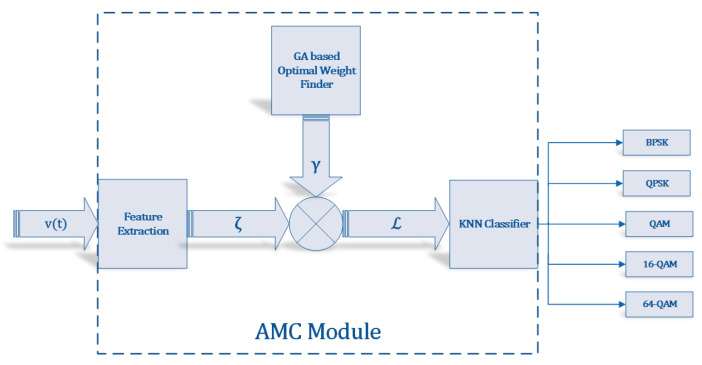
Proposed AMR module.

**Figure 3 sensors-22-07488-f003:**
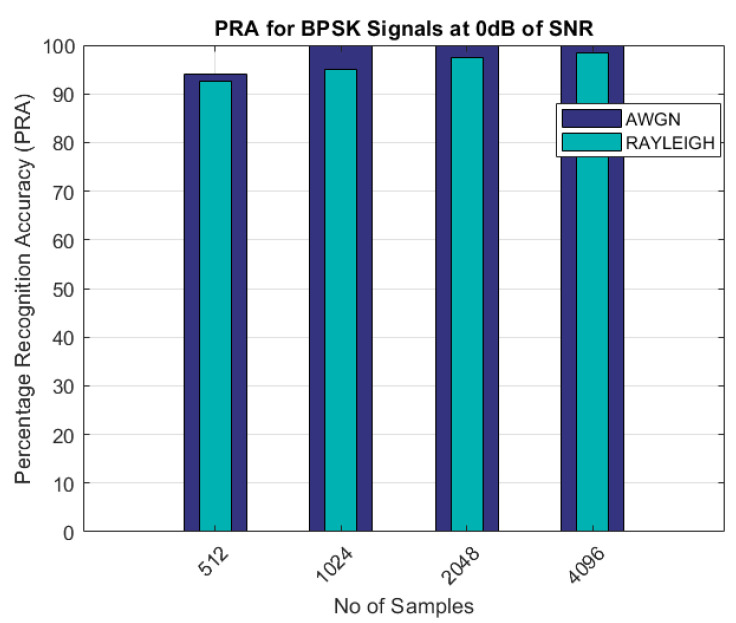
PRA for BPSK signals.

**Figure 4 sensors-22-07488-f004:**
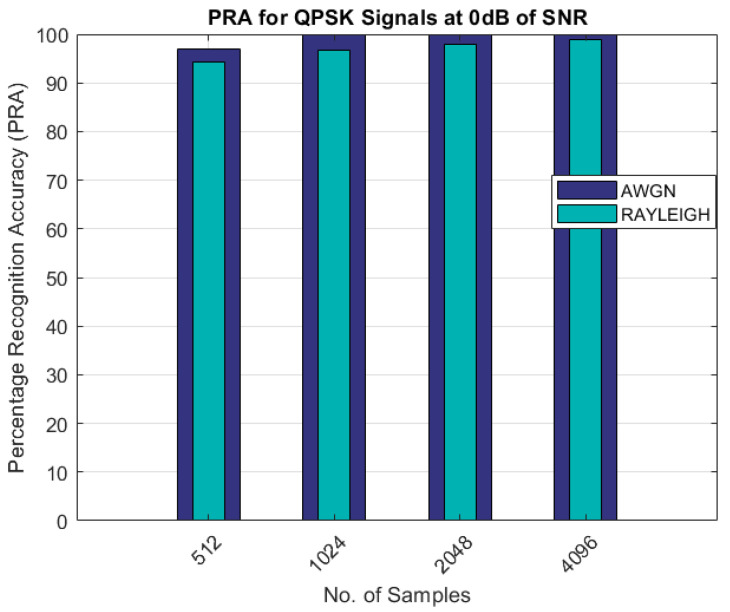
PRA for QPSK signals.

**Figure 5 sensors-22-07488-f005:**
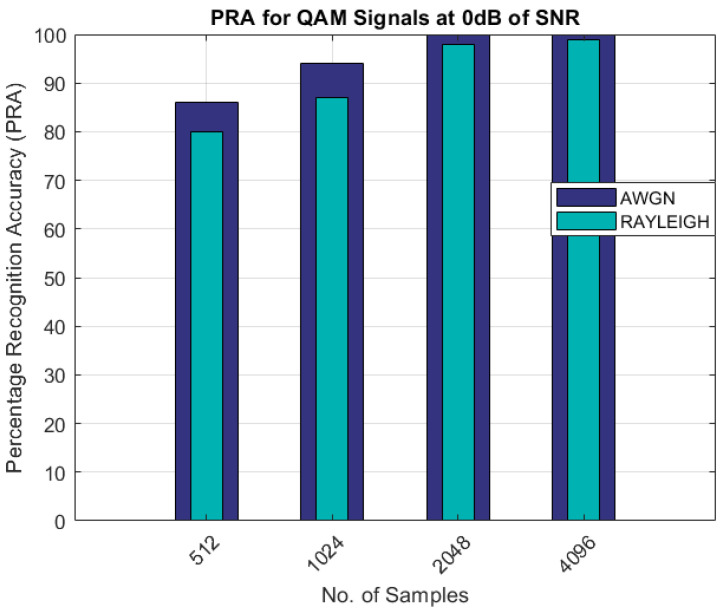
PRA for QAM signals.

**Figure 6 sensors-22-07488-f006:**
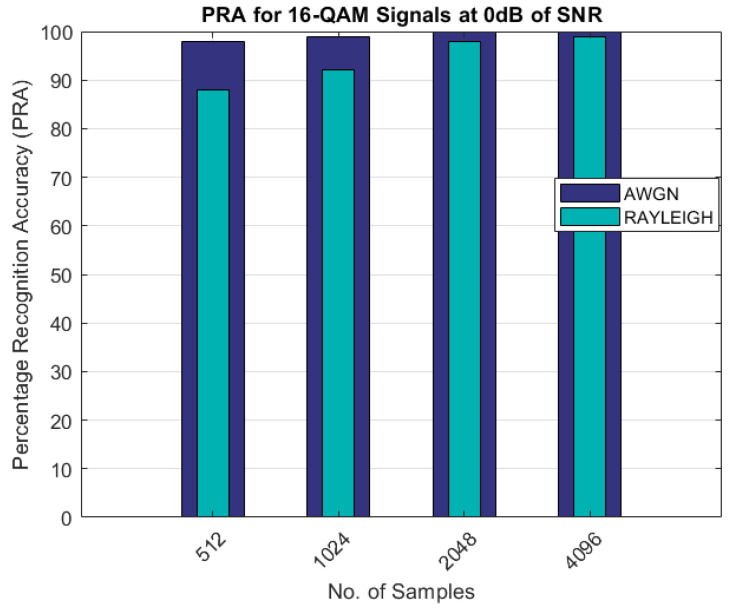
PRA for 16-QAM signals.

**Figure 7 sensors-22-07488-f007:**
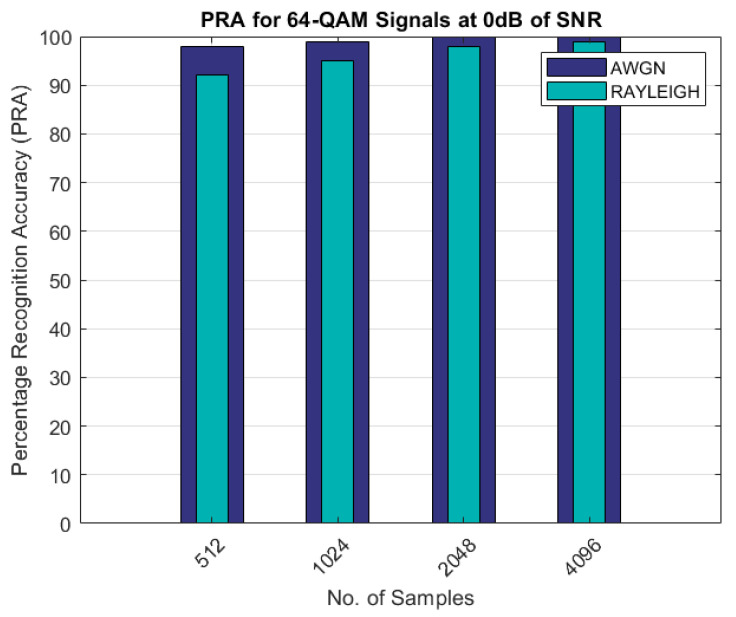
PRA for 64-QAM signals.

**Table 1 sensors-22-07488-t001:** HOC values.

	$\zeta_{1}$	$\zeta_{2}$	$\zeta_{3}$	$\zeta_{4}$	$\zeta_{5}$	$\zeta_{6}$	$\zeta_{7}$	$\zeta_{8}$	$\zeta_{9}$	$\zeta_{10}$
**BPSK**	0.78	2.45	0.29	0.27	0.46	31.41	19.65	77.40	123.07	2284.53
**QPSK**	0.05	2.41	0.09	0.03	0.58	32.95	0.40	77.87	12.49	2210.43
**QAM**	0.16	3.35	0.03	0.28	0.40	0.83	42.92	210.23	1.27	8477.33
**16-QAM**	0.79	11.64	0.10	0.51	0.56	13.34	384.43	956.81	331.22	13,052.11
**64-QAM**	1.10	42.16	0.04	0.54	0.58	31.43	1004.16	4053.52	1186.00	228,652.61

**Table 2 sensors-22-07488-t002:** LC of HOC’s.

Linear Combinations	BPSK	QPSK	QAM	16-QAM	64-QAM
	2372.2	2211.3	8307.9	12854.9	218736.5

**Table 3 sensors-22-07488-t003:** Simulation parameters.

Parameter	Standard Value
No. of Samples	[512, 1024, 2048, 4096]
SNR	[0–5] dB
Training of Recognizer	70%
Testing of Recognizer	20%
No. of Genes	120
No. of Chromosomes	1024
Crossover Fraction	0.25
Crossover	Heuristic
Selection	Stochastic Uniform
Mutation	Adaptive Feasible
Elite Count	2

**Table 4 sensors-22-07488-t004:** Percentage recognition accuracy on the AWGN channel.

PRA for BPSK			
**No. of Samples**	**0 dB**	**5 dB**	**10 dB**
512	90	99.01	100
1024	100	100	100
2048	100	100	100
4096	100	100	100
**PRA for QPSK**			
**No. of Samples**	**0 dB**	**5 dB**	**10 dB**
512	97	100	100
1024	99.90	100	100
2048	100	100	100
4096	100	100	100
**PRA for QAM**			
**No. of Samples**	**0 dB**	**5 dB**	**10 dB**
512	86	99	100
1024	94	99.99	100
2048	100	100	100
4096	100	100	100
**PRA of 16-QAM**			
**No. of Samples**	**0 dB**	**5 dB**	**10 dB**
512	98	99.98	100
1024	99.99	100	100
2048	100	100	100
4096	100	100	100
**No. of Samples**	**0 dB**	**5 dB**	**10 dB**
512	98	99	100
1024	99.95	100	100
2048	100	100	100
4096	100	100	100

**Table 5 sensors-22-07488-t005:** Percentage recognition accuracy on the Rayleigh channel.

PRA for BPSK			
**No. of Samples**	**0 dB**	**5 dB**	**10 dB**
512	92.5	94.7	96.1
1024	95	97.2	98
2048	97.5	98.2	99
4096	98.5	99.5	100
**PRA for QPSK**			
**No. of Samples**	**0 dB**	**5 dB**	**10 dB**
512	94.2	96.5	97
1024	96.7	98	99.5
2048	98	99	100
4096	99	100	100
**PRA for QAM**			
**No. of Samples**	**0 dB**	**5 dB**	**10 dB**
512	80	88	96
1024	87	93	97
2048	98	99	100
4096	99.5	100	100
**PRA for 16-QAM**			
**No. of Samples**	**0 dB**	**5 dB**	**10 dB**
512	88	95	97
1024	92	97	99
2048	98	98.5	100
4096	99	99.7	100
**PRA for 64-QAM**			
**No. of Samples**	**0 dB**	**5 dB**	**10 dB**
512	92	96	99
1024	95	96	99
2048	98	98.5	100
4096	99	100	100

**Table 6 sensors-22-07488-t006:** PRA comparison on the AWGN channel model.

Modulation Schemes	Keshk et al. [60]		Ali et al. [36]		Chen et al. [39]		Ghauri et al. [64]		Hussain et al. [54]		Proposed Classifier	
**No. of Samples**	**–**		**–**		**–**		**2048 Samples**		**1024 Samples**		**1024 Samples**	
**SNR**	**0 dB**	**5 dB**	**0 dB**	**5 dB**	**0 dB**	**8 dB**	**0 dB**	**5 dB**	**0 dB**	**5 dB**	**0 dB**	**5 dB**
BPSK	50	65	-	98	-	99	-	-	98	99.9	100	100
QPSK	73	86	-	98	-	98	-	-	99.9	100	99.9	100
QAM	-	-	96	-	-	-	72	98	91	99	94	99.9
16-QAM	-	-	97	-	-	97	72	97	99.8	99.9	99.9	100
64-QAM	-	-	98	-	-	97	70	98	99	99.9	99.9	100

**Table 7 sensors-22-07488-t007:** PRA comparison on the Rayleigh channel model.

Modulation Schemes	No. of Samples (512)				No. of Samples (1024)			
	SNR							
	0 dB	0 dB	5 dB	5 dB	0 dB	0 dB	5 dB	5 dB
	[54]	Proposed	[54]	Proposed	[54]	Proposed	[54]	Proposed
**BPSK**	42	**92.5**	56	**94.7**	68	**95**	70	**97.2**
**QPSK**	41	**94.2**	48	**96.5**	45	**96.7**	68	**98**
**QAM**	30	**80**	41	**88**	45	**89**	50	**93**
**16-QAM**	35	**88**	53	**95**	43	**92**	58	**97**
**64-QAM**	47	**92**	58	**96**	51	**95**	60	**96**

## Data Availability

Not applicable.

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
