# Peer review of "Automatic Modulation Recognition Based on the Optimized Linear Combination of Higher-Order Cumulants"

_sensors, 2022, doi:10.3390/s22197488_

Round 1

Reviewer 1 Report

Authors provided clear motivation and contribution of the proposed study. The whole manuscript is well-structed and readable.  

The presented algorithm shows high accuracy for the modulation schemes even at lower SNRs during simulation scenarios. I would expect to see furthermore the performance of algorithm on application scenarios. 

Author Response

Thankyou so much for highlighting issues in our manuscript and we have updated the file.

Reviewer 2 Report

Dear Authors,

I think the paper in general is good, but some comments need to be considered to improve the content of the paper. The following are my comments:

1- The introduction needs to be improved to cover the topics in this paper.

2- Related works section is perfect.

3- show samples in plots for each phase of the system like parameter Extraction (HOC), ...etc.

4- You mentioned that data is separated into 70% for training and 20% for testing. Where is the missed 10% is it used for evaluation?

5- GA based Optimal Weight Finder algorithm error plot with generations or iterations.

6- show a scatter plot for extracted features to see If there can discrimination between different classes.

7- what is the setting for the KNN classifier?

8- Show a detailed Proposed System Model, the recognized is missed in the graph.

Author Response

Thankyou for kind suggestions to update the manuscript. We have properly update the manuscript.
